

# Transcriptomic and metabolomic analysis of autumn leaf color change in *Fraxinus angustifolia*

Yanlong Wang[1,*], Jinpeng Zhen[2,*], Xiaoyu Che[1], Kang Zhang[2], Guowei Zhang[3], Huijuan Yang[1], Jing Wen[1], Jinxin Wang[1], Jiming Wang[1,4], Bo He[5], Ailong Yu[6], Yanhui Li[1] and Zhigang Wang[1]

[1] College of Landscape Architecture and Tourism, Hebei Agricultural University, Baoding, China
[2] Hebei Key Laboratory of Plant Physiology and Molecular Pathology, Hebei Bioinformatics Utilization and Technological Innovation Center for Agricultural Microbes, Hebei Agricultural University, Baoding, China
[3] Hongyashan State-owned Forest Farm in Hebei Province, Baoding, China
[4] College of Grammar, Hebei Normal University of Science and Technology, Qinhuangdao, China
[5] Green Building Development Center of Baoding, Baoding, China
[6] Flower and Wood Technical Service Center of Hengshui, Hengshui, China
[*] These authors contributed equally to this work.

Corresponding authors
Yanhui Li, yanhuili01@163.com
Zhigang Wang, wzhg1956@163.com

## ABSTRACT

*Fraxinus angustifolia* is a type of street tree and shade tree with ornamental value. It has a beautiful shape and yellow or reddish purple autumn leaves, but its leaf color formation mechanism and molecular regulation network need to be studied. In this study, we integrated the metabolomes and transcriptomes of stage 1 (green leaf) and stage 2 (red-purple leaf) leaves at two different developmental stages to screen differential candidate genes and metabolites related to leaf color variation. The results of stage 1 and stage 2 transcriptome analysis showed that a total of 5,827 genes were differentially expressed, including 2,249 upregulated genes and 3,578 downregulated genes. Through functional enrichment analysis of differentially expressed genes, we found that they were involved in flavonoid biosynthesis, phenylpropanoid biosynthesis, pigment metabolism, carotene metabolism, terpenoid biosynthesis, secondary metabolite biosynthesis, pigment accumulation, and other biological processes. By measuring the metabolites of *Fraxinus angustifolia* leaves, we found the metabolites closely related to the differentially expressed genes in two different periods of *Fraxinus angustifolia*, among which flavonoid compounds were the main differential metabolites. Through transcriptome and metabolomics data association analysis, we screened nine differentially expressed genes related to anthocyanins. Transcriptome and qRT-PCR results showed that these nine genes showed significant expression differences in different stages of the sample, and we speculate that they are likely to be the main regulatory factors in the molecular mechanism of leaf coloration. This is the first time that we have analyzed the transcriptome combination metabolome in the process of leaf coloration of *Fraxinus angustifolia*, which has important guiding significance for directional breeding of colored-leaf *Fraxinus* species and will also give new insights for enriching the landscape.

## INTRODUCTION

Colored-leaf plants refer to plants whose leaves are stable, non-green, and of obvious ornamental value in certain seasons or throughout the year (*Wang et al., 2016*). As important as flower color, leaf color is a significant ornamental trait of plants. The pigment content in colored-leaf plants is affected by specific factors, especially the ratio of carotenoids or anthocyanins to chlorophyll, which leads to red or yellow leaves (*Cheng et al., 2015*). Developed countries have been engaged in the cultivation and application of colorful plants for more than one 100 years. In the United States, Canada, and other countries, the planting area of colored-leaf trees can reach more than 30% that of all trees (*Özbayram & Çiçek, 2018*).

There are three main categories of pigments: chlorophyll, mainly chlorophyll a and chlorophyll b; carotenoids, mainly carotenoids and lutein; and three flavonoid pigments, also known as anthocyanins (*Tanaka, Sasaki & Ohmiya, 2008*). Different pigments appear in different colors; ordinary leaves contain more chlorophyll than carotenoids, so the leaves are always green (*Tanaka, Sasaki & Ohmiya, 2008*). Studies have shown that a combination of flavonoids, anthocyanins, and carotenoids can increase leaf color diversity (*Luiza Koop et al., 2022*). *Ginkgo biloba* L, *Pseudolarix amabilis,* and other leaves contain more carotenoids and appear yellow; anthocyanins, such as in *Acer palmatum* Thunb and *Acer buergerianum* Miq, appear red when their content increases. *Amaranthus tricolor* has leaves from purple to bright red, which is also due to cell senescence and chlorophyll reduction, so that the color of anthocyanins shows (*Sharma, Mazumdar & Keshav, 2021*). Carotenoids and anthocyanins also play an important role in the photoprotection of photosystems, which help plants adapt to the environment and maintain the proper functioning of their internal mechanisms during leaf senescence (*Das et al., 2011*; *Hashimoto, Uragami & Cogdell, 2016*).

The massive accumulation of anthocyanins is the basic material for the reddening of colorful plant leaves (*Zhang et al., 2019*). Many studies have shown that the change in leaf color anthocyanin content in colored-leaf plants is the result of interaction of environmental factors and internal factors (*Santos-Buelga, Mateus & De Freitas, 2014*). Temperature drop and a shortened photoperiod in autumn and winter often trigger genetic signals for chlorophyll decomposition (*Song, Ito & Imaizumi, 2013*). With the decomposition of amino acids, proteins, and other substances in leaves and migration of nutrients in leaves, the chlorophyll content gradually decreases with time, while the anthocyanin content gradually increases, making the leaves red (*Li et al., 2018*). At the same time, the synthesis and accumulation of anthocyanins is closely related to sugar, protein, chlorophyll, carotenoids, and other substances in plants, and is affected by natural environmental factors such as light time, light intensity, and temperature changes (*Zhang et al., 2016*).

Anthocyanins are produced by flavonoid compounds through the shikimic acid synthesis pathway, synthesized in the cytoplasm, and eventually stored in plant cell vacuoles (*Mishio, Takeda & Iwashina, 2015*). This synthetic pathway is an important branch of the plant flavonoid biosynthesis pathway, and it is also one of the most important pathways for the synthesis of plant secondary metabolites (*Zhao et al., 2022*). The precursors of

anthocyanin synthesis are malonic acid monophthalide-CoA and 4-hydroxybenzoic acid-CoA, and the final product anthocyanin is produced by a series of key enzymes, such as chalcone synthase (CHS), chaleone isomerase (CHI), dihydroflavonol reductase (DFR), and glycosyltransferase (GT) (*Tanaka, Sasaki & Ohmiya, 2008*). Anthocyanin metabolism is affected by key enzyme genes and transcription factors (*Tanaka, Sasaki & Ohmiya, 2008*). Key enzyme genes directly encode the key enzymes required for anthocyanin synthesis (*Zhou et al., 2017*).

*Fraxinus angustifolia* is an important afforestation tree species in northern China (*Sarfraz et al., 2017*). *Fraxinus angustifolia* has an upright body, straight trunk, and luxuriant branches and leaves, and it is an excellent street tree and shade tree, as autumn-colored tree species have become favored for good landscape effects in recent years (*Manzanera & Martinez-Chacon, 2007*; *Özbayram & Çiçek, 2018*). However, research on the change of leaf color has been limited. In this study, transcriptome and metabolome sequencing analysis was performed on the leaves of two critical periods of *Fraxinus angustifolia* to explore the genes related to the leaf coloration of *Fraxinus angustifolia* and their regulatory networks, and then to screen out the genes that may be related to the metabolic process of anthocyanins and leaf color expression. The results of this study provide a theoretical basis for elucidating the mechanism of leaf color formation, and they reveal the role and molecular mechanism of anthocyanins in the leaves of *Fraxinus angustifolia*. This has important guiding significance for directional cultivation of colored-leaf *Fraxinus* varieties and will also provide important technical support for the study of molecular regulation mechanisms of colored-leaf traits in woody plants.

## MATERIALS AND METHODS

### Plant materials

The leaves of 6-year-old *Fraxinus angustifolia* were collected from Baoding, Hebei Province in October. Then, the leaves of *Fraxinus angustifolia* were quickly frozen in liquid nitrogen bottles, and then transferred to an ultra-low temperature refrigerator for cryopreservation for RNA and metabolite extraction.

### RNA extraction, library construction, and sequencing

Total RNA was extracted from the sample using TRIzol (Invitrogen, Waltham, MA, USA) and purified using a RNeasy column (Qiagen, Hilden, Germany). mRNAs were enriched with Oligo(dT) beads After total RNA extraction. Enriched mRNA fragments were split into short fragments and reverse transcribed into cDNA. The cDNA fragment was then purified using the QiaQuick PCR extraction kit (Qiagen, Hilden, Germany), the ends were repaired, poly(A) was added, and it was connected to the Illumina sequencing adapter. The ligation products were size selected by agarose gel electrophoresis. After PCR amplification and quality inspection, DNA fragments were sequenced using Illumina HiSeq™ 4000.

### Transcriptome low quality data filtering

FastQC (*Brown, Pirrung & McCue, 2017*) software was used to filter the data. The parameters were as follows: (a) delete the read containing the adapter; (b) remove reads

containing more than 10% unknown nucleotides (N); (c) remove low quality reads containing more than 50% low quality ($q$ value $\leq 20$) bases.

## Transcriptome data assembly and integrity assessment

The obtained high-quality reads were assembled and annotated using Trinity software (*Sewe et al., 2022*). Trinity is a modular approach and software package that consists of three parts: Inchworm, Chrysalis, and Butterfly. First, the Inchworm was read by the k-mer method to obtain a linear contigs set. Next, Butterfly aggregated the related contigs corresponding to the unique part of the alternatively spliced transcript or the accessory gene and then constructed a de Bruijn diagram for each related contigs cluster. Finally, Butterfly analyzed the path taken to read and read the pairing in the corresponding de Bruijn diagram, and it then output a linear sequence for each alternatively spliced isoform and script obtained from the fallacy gene. At the same time, BUSCO software was used to evaluate the integrity of the assembly results (*Manni et al., 2021*).

## Transcriptome annotation and differential gene analysis

Firstly, the assembled Unigene sequences were compared to the Nr database, KEGG, COG/KOG, and SwissProt by blastx. Based on Nr annotation information, we used Blast2GO software to obtain GO function annotation (*Conesa & Gotz, 2008*). At the same time, the Pfam_Scan program was used to compare with the Pfam database to obtain annotation information related to protein structure. The HMMER program was compared with the SMART database to obtain the annotation information of the protein domain. The PlantTFDB database was compared by hmmscan to obtain transcription factor annotation information (*Jin et al., 2017*). DESeq2 software was used to analyze differentially expressed genes (DEGs) between groups (*Love, Huber & Anders, 2014*). The analysis was divided into three parts: (a) standardization of readcount; (b) calculation of the probability of hypothesis testing ($P$-value) based on the model; and (c) multiple hypothesis test correction to obtain the FDR value (false discovery rate). Based on the results of differential analysis, the genes of FDR were screened as significantly differential genes.

## Gene function enrichment analysis

Gene ontology is an internationally standardized gene function classification system that fully describes the characteristics of genes and their products in organisms. The differential genes are mapped to GO database. AgriGO v2.0 custom tools KOBAS-i was employed to GO and KEGG enrichment analysis, respectively (*Tian et al., 2017*; *Bu et al., 2021*).

## Metabolite extraction

The biomaterial samples, preserved at ultra-low temperatures, were taken out and vacuum freeze-dried. Samples were ground by a grinding machine at 30 Hz for 1.5 min, and 100 mg of powder was weighed. The powder was extracted overnight at 4 °C with 70% methanol 1.0 ml containing 0.1 mg/l lidocaine as an internal standard, and vortexed three times to make the extraction more sufficient. After extraction, it was centrifuged at 10,000 g for 10 min, the supernatant was aspirated, and the sample was filtered through a microporous membrane (0.22 μm pore size) and stored in a sample bottle for subsequent LC-MS

analysis. A quality control sample (QC) was prepared by mixing sample extracts to analyze the repeatability of samples under the same treatment method. Usually, a QC sample was inserted every 10 samples to investigate the repeatability of the analysis process.

## Detection of metabolites

Conditions for UPLC analysis were as follows: An analytical column mobile phase: ultrapure water (0.04% acetic acid added), acetonitrile (0.04% acetic acid added); elution gradient: 0 min water/acetonitrile (95:5V/V), 11.0 min 5:95V/V, 12.0 min 5:95V/V, 12.1 min 95:5V/V, and 15.0 min 95:5V/V. Waters ACQUITY UPLC HSS T3 C18 1.8. There was a 0.4 ml/min flow; 40 °C for the column; a 2 l input volume. ESI-QTRAP-MS was used to evaluate the separated samples.

The primary settings for the linear ion trap and triple quadrupole in the API 6500 QTRAP LC/MS/MS system were as follows: collision-activated dissociation (CD) parameter setting of high, electrospray ionization (ESI) temperature of 550 ° C, mass spectrometry voltage of 5,500 V, curtain gas (CUR) of 25 psi. Each ion pair is scanned in a triple quadrupole (QQQ) using an optimum declustering potential (DP) and collision energy (CE). The software Analyst 1.6.1 was used to process the data.

## Metabolomics data analysis

Peaks were manually examined for signal/noise (s/n) >10, and internal Perl software was used to eliminate duplicate signals brought on by various isotopes, in-source fragmentation, $K^+$, $Na^+$, and $NH_4^+$ adduct, and dimerization. This resulted in a matrix with fewer skewed and redundant data. Accurate m/z for each Q1 was collected to make it easier to identify and annotate metabolites. To provide a summary of the metabolite profiles of all samples, total ion chromatograms (TICs) and extracted ion chromatograms (EICs or XICs) of QC samples were exported. Each chromatographic peak's area was determined. Peak alignment was done between the several samples using the spectral pattern and retention time. Metabolites were identified by searching internal database and public databases (MassBank, KNApSAcK, HMDB, MoToDB, and METLIN) and comparing the m/z values, the RT, and the fragmentation patterns with the standards.

Statistical analysis (PCA analysis and OPLS-DA analysis) used the VIP value and *P* value to screen differential metabolites. Through inter-group difference analysis, differentially expressed genes were obtained from transcriptome data, differentially expressed metabolites were obtained from metabolomics data, and KEGG enrichment analysis of each group was performed. The correlation analysis of common KEGG pathways was carried out for the differential genes and metabolites between groups.

## Transcriptome metabolome O2PLS model analysis

Based on all transcriptome and all metabolome data, we performed O2PLS analysis using the OmicsPLS package (*Bouhaddani et al., 2018*). The O2PLS model is suitable for association analysis of two omics data and can be used for bidirectional modeling and prediction in two data matrices. Through calculation, the O2PLS model decomposes the data of each omics into three parts, namely, the joint part (a part that is strongly related to another group), the orthogonal part (parts that affect only one group of data and have no
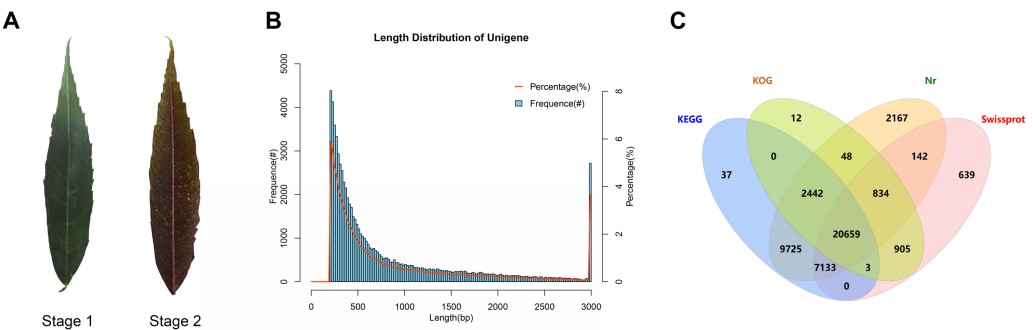

**Figure 1** **Statistical analysis of transcriptome data.** (A) Leaf states at stage 1 and stage 2. (B) Length distribution of Unigene in transcriptome assembly. (C) Venn diagram of Unigene annotation in the Nr, KEGG, KOG, and SwissProt databases.

effect on another omics data), and the noise part (parts that have no effect on both sets of data). The degree to which each component explains the total variation is expressed as R2, and Higher values represent better model explanatory power. Overfitting and underfitting of models reduce R2.

## Correlation coefficient analysis

The Pearson correlation coefficient can be used to measure the relationship between two variables, representing the strength of co-variation between two variables, ranging from −1 to +1. The Pearson coefficient of the expression of all differential genes (the union of differential genes between each comparison group) and the abundance of all differential metabolites (the union of differential metabolites between each comparison group) was calculated to evaluate the correlation between genes and metabolites.

# RESULTS

## Transcriptome data filtering and assembly analysis

To explore the mechanism of leaf color change in *Fraxinus angustifolia*, we sequenced the *de novo* transcriptomes of *Fraxinus angustifolia* leaves at two different growth stages. Based on the transcriptome data, after quality control, the clean reads of six samples were 5,766,222,300–10,874,368,200, the Q30 base percentage was 93.73–95.20%, and the GC content was 44.98–45.44% (Table S1). The overall sequencing filtering quality was good and could be used for subsequent transcriptome analysis. Through transcriptome assembly, we obtained 74,430 Unigenes with a length ranging from 201 bp to 16,676 bp, with an average length of 863 bp (Fig. 1B). According to the distribution of Unigene with different lengths after assembly, the length of 0–500 bp accounted for about 60% of the total. In the splicing results, the N50 length was 1,458 bp. The integrity of the assembled transcripts was evaluated by BUSCO (Fig. S1), and the integrity was 75.56%, indicating that the integrity of the transcriptome data was high.
## Functional annotation analysis of unigenes

To explore the gene function of the unigenes, we used four databases (Nr, KEGG, KOG, SwissProt) to annotate the obtained Unigenes (Fig. 1C). Finally, 44,746 genes with annotation information were obtained, and 29,684 Unigenes were not annotated, with an annotation ratio of 60.12%. Among them, 43,150 Unigenes were annotated in the Nr database, 39,999 Unigenes were annotated in the KEGG database, 24,903 Unigenes were annotated in the KOG database, and 30,315 Unigenes were annotated in the Swissprot database. Among them, 20,659 genes had annotation information in four databases.

## Gene differential expression analysis of transcriptome data

In order to ensure the accuracy of subsequent analysis, we first corrected the sequencing depth, and then corrected the length of the gene or transcript to obtain the RPKM value of the gene, and then performed the subsequent analysis. Based on the RPKM value of each gene, the expression distribution was calculated. As can be seen from the distribution, gene expression is more uniform between different samples, meaning that these results can be used for subsequent differential gene analysis (Fig. 2A). Principal component analysis (PCA) and pearson correlation coefficient analysis clustered the samples based on the expression level. Through PCA analysis between these samples, results showed that three biological replicates of samples from different periods were distributed in the same quadrant (Fig. 2B). Pearson correlation coefficient analysis showed the correlation of each two samples was above 0.85 (Fig. S2), indicating that the sample repeatability was good. A total of 5,827 DEGs were found between stage 1 (green leaf stage) and stage 2 (red purple leaf stage), of which 2,249 were upregulated and 3,578 were downregulated (Figs. 2C and 2D).

## Functional analysis of differential genes

To elucidate the function of differentially expressed genes in leaf color change of *Fraxinus angustifolia*, GO enrichment analysis of differentially expressed genes was performed. Based on the results of GO functional enrichment analysis, the differentially expressed genes in stage 1 (green leaf stage) and stage 2 (red purple leaf stage) were mainly enriched in flavonoid anabolism, programmed cell death, phenylpropanoid anabolism, chloroplast quinone metabolism, pigment metabolism, carotene metabolism, polysaccharide metabolism, terpenoid anabolism, jasmonic acid metabolism, fatty acid anabolism, photosynthesis, secondary metabolite biosynthesis, pigment accumulation, and other biological processes (Fig. 3A). In the molecular function classification, the differentially expressed genes were significantly enriched in terms of oxidoreductase activity, monooxygenase activity, hydrolase activity, transferase activity, tetrapyrrole binding, nucleic acid binding transcription factor activity, carotenoid dioxygenase activity, antioxidant activity, fatty acid synthase activity, galactosyltransferase activity, and iron ion binding (Fig. 3B). In the classification of cell composition, GO terms were mainly enriched in the thylakoid, chloroplasts, photosynthetic membranes, cell wall, peroxisome, photosystem, light-harvesting complex, thylakoid membrane, *etc* (Fig. 3C).

Then, KEGG pathway enrichment analysis was performed with differentially expressed genes to analyze the key enriched metabolic pathways in leaf color change of *Fraxinus*
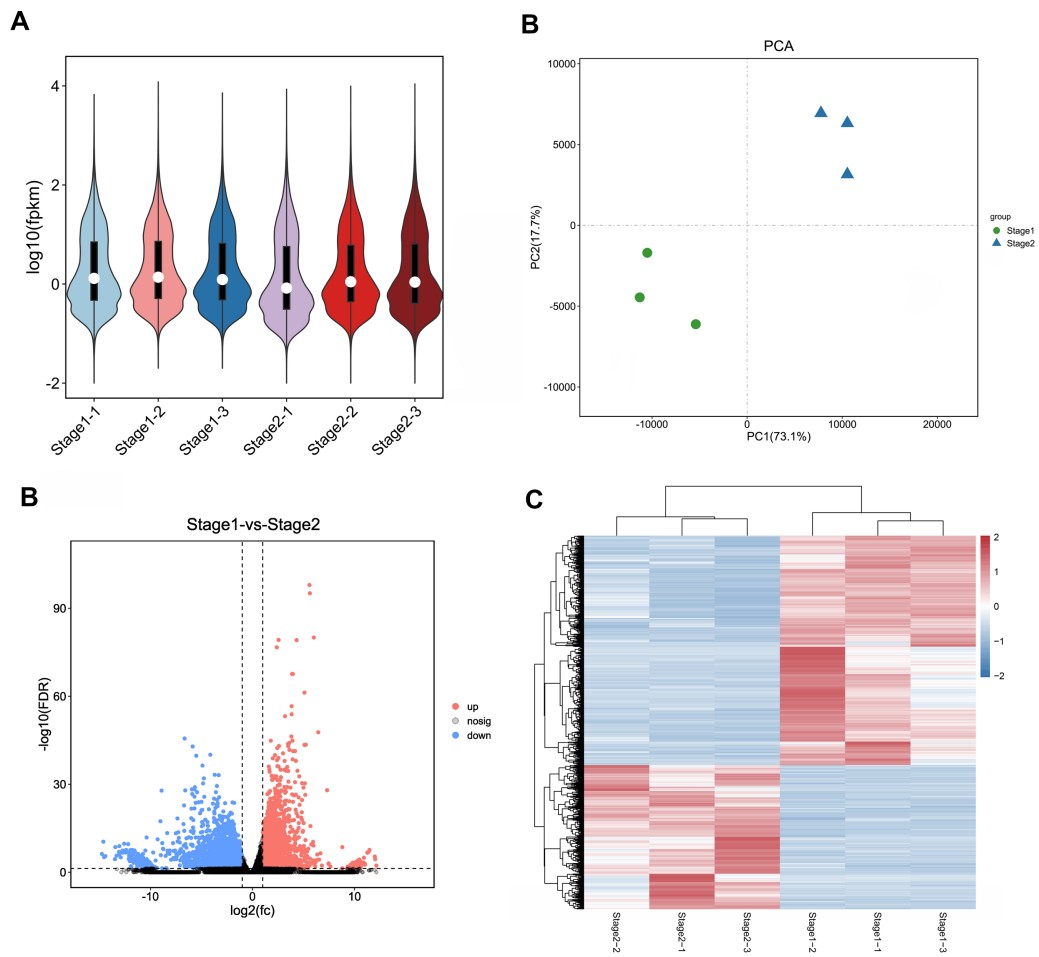

**Figure 2  Expression analysis of Unigenes in stage 1 and stage 2.** (A) Expression level analysis of different biological replicates in stage 1 and stage 2. (B) Principal component analysis (PCA) of expression datasets in stage 1 and stage 2. (C) Volcano-plots of differentially expressed genes in stage 1 and stage 2. (D) Heatmap of differentially expressed genes in stage 1 and stage 2. The numbers ranging from −2 to 2 represented the expression level from low to high based $Z$-score method.

*angustifolia.* Based on the results of KEGG metabolic pathway enrichment analysis, we found that the differentially expressed genes between stage 1 (green leaf) and stage 2 (red-purple leaf) were significantly enriched in metabolic pathways such as secondary metabolite biosynthesis, phenylpropanoid biosynthesis, riboflavin metabolism, flavonoid biosynthesis, anthocyanin biosynthesis, carotene biosynthesis, terpenoid biosynthesis, steroid biosynthesis, and thiamine metabolism (Fig. 4). The functional enrichment analysis indicated that differentially expressed genes played an important role in leaf color change of *Fraxinus angustifolia.*

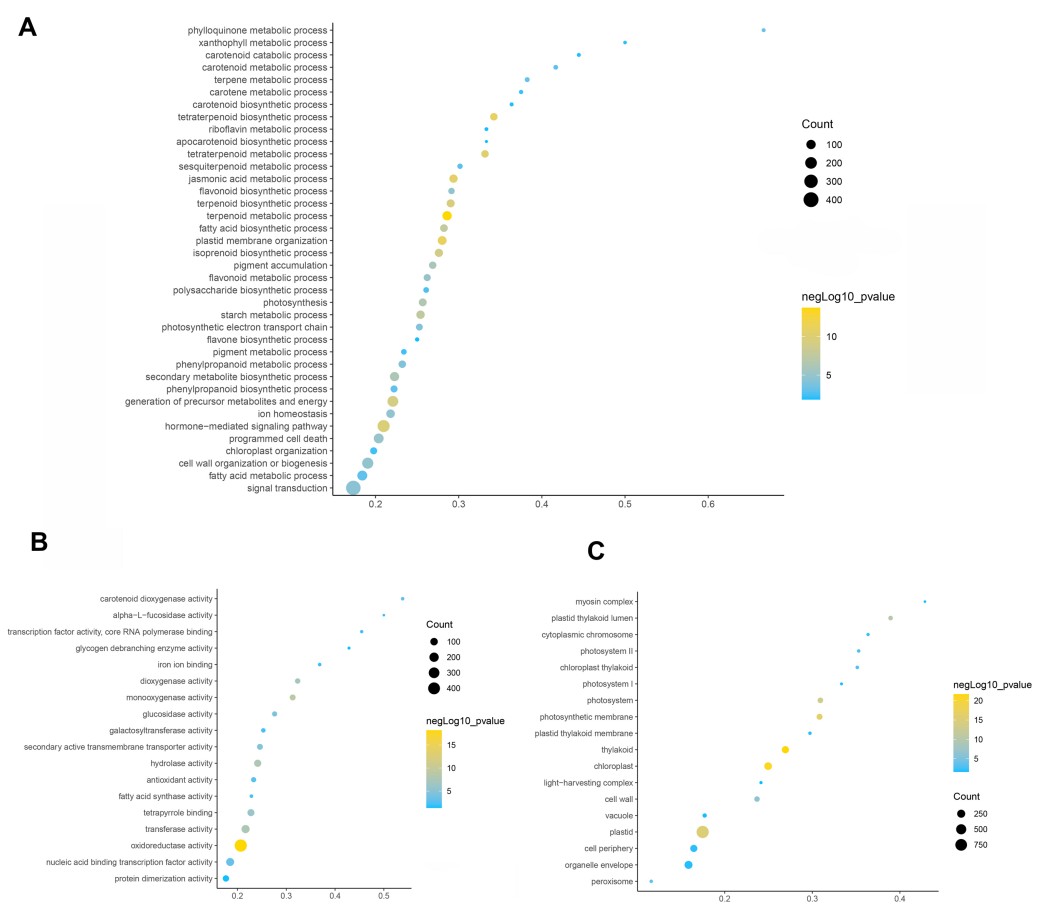

**Figure 3** **GO function enrichment analysis of differentially expressed genes.** (A) Biological progress. (B) Molecular function. (C) Cellular component. Fisher's exact test, adjusted *P*-value was performed by the Benjamini–Yekutieli method (FDR < 0.05).

## Analysis of differential metabolites in leaves of *Fraxinus angustifolia* at different stages

Leaves show different colors mainly due to internal pigment-related components and content. The leaves of different periods of *Fraxinus angustifolia* have significant color differences in phenotype (Fig. 1A). Through the determination of the metabolome, the leaves of *Fraxinus angustifolia* in these two periods can be divided into two groups with significant differences according to PCA analysis, based on the content of their internal metabolites (Fig. 5A). To further identify the metabolites associated with leaf color difference, we used orthogonal partial least squares discriminant analysis (OPLS-DA) to explore the differential metabolites between different colors.

Cross-validation and ranking validation showed that the OPLS-DA model of each comparison group was meaningful (Fig. 5B). Then, we obtained the relevant load diagram by OPLS-DA analysis (Fig. 5B and Fig. S3). The farther the metabolites from the origin in the OPLS-DA loading plot, the greater their contribution to the separation of leaf color categories of *Fraxinus angustifolia*. Organic acids, lipids, and flavonols are the main

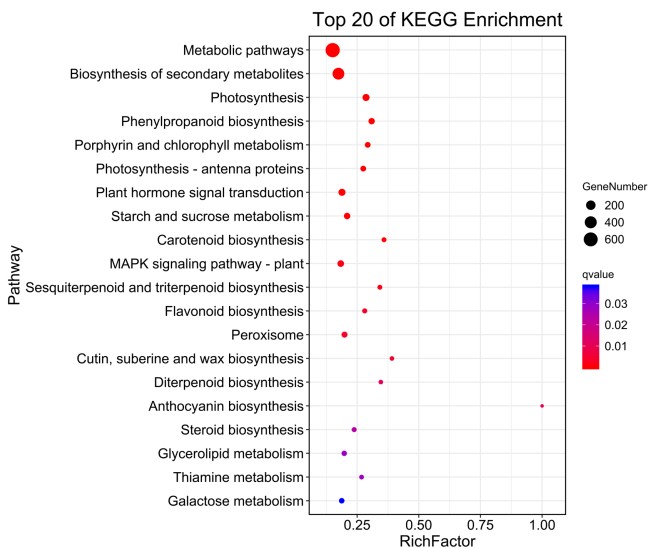

**Figure 4   KEGG enrichment analysis of differentially expressed genes of stage 1 *vs* stage 2.** The abscissa indicates the degree of significant enrichment.

substances that cause differences in the metabolome level of *Fraxinus angustifolia* at different stages, while flavonols are the main factors that cause differences in leaf color at different stages.

Furthermore, according to the criteria for identifying differential metabolites (VIP>1, $t$-test $P$-value<0.05), we identified a total of 118 compounds. Compared with stage 1 (green leaf stage) and stage 2 (red purple leaf stage), with 40 kinds of material accumulation, there were 13 kinds of flavonoid compounds, including eight kinds of flavonols and four kinds of flavonoids (Fig. 5C). Therefore, we speculate that flavonoid compounds may be the main factor causing the change of leaf color of *Fraxinus angustifolia*.

Based on the metabolic pathway enrichment analysis of differential metabolites, we found that the differential metabolites in stage 1 (green leaf stage) and stage 2 (red purple leaf stage) were mainly enriched in secondary metabolite biosynthesis, aminoacyl-tRNA biosynthesis, 2-oxocarboxylic acid metabolism, glucosinolate biosynthesis, carbapenem biosynthesis, cyanoamino acid metabolism, phenylpropanoid biosynthesis, ubiquinone and other terpenoid quinone biosynthesis, riboflavin metabolism, porphyrin and chlorophyll metabolism, fatty acid biosynthesis, the TCA cycle, tryptophan metabolism, niacin and nicotinamide metabolism, and other metabolic pathways (Fig. 5D). Based on the above results, we found that the pathways related to anthocyanin synthesis were significantly enriched, which provided an important basis for subsequent analysis.

## Association analysis of DEGs and metabolites between the green and red-purple leaf stage

To calculate the gene expression and metabolite abundance, we used Pearson correlation coefficient to evaluate the correlation between genes and metabolites in stage 1 and stage 2. Then, we performed a loading plot analysis on the association between genes

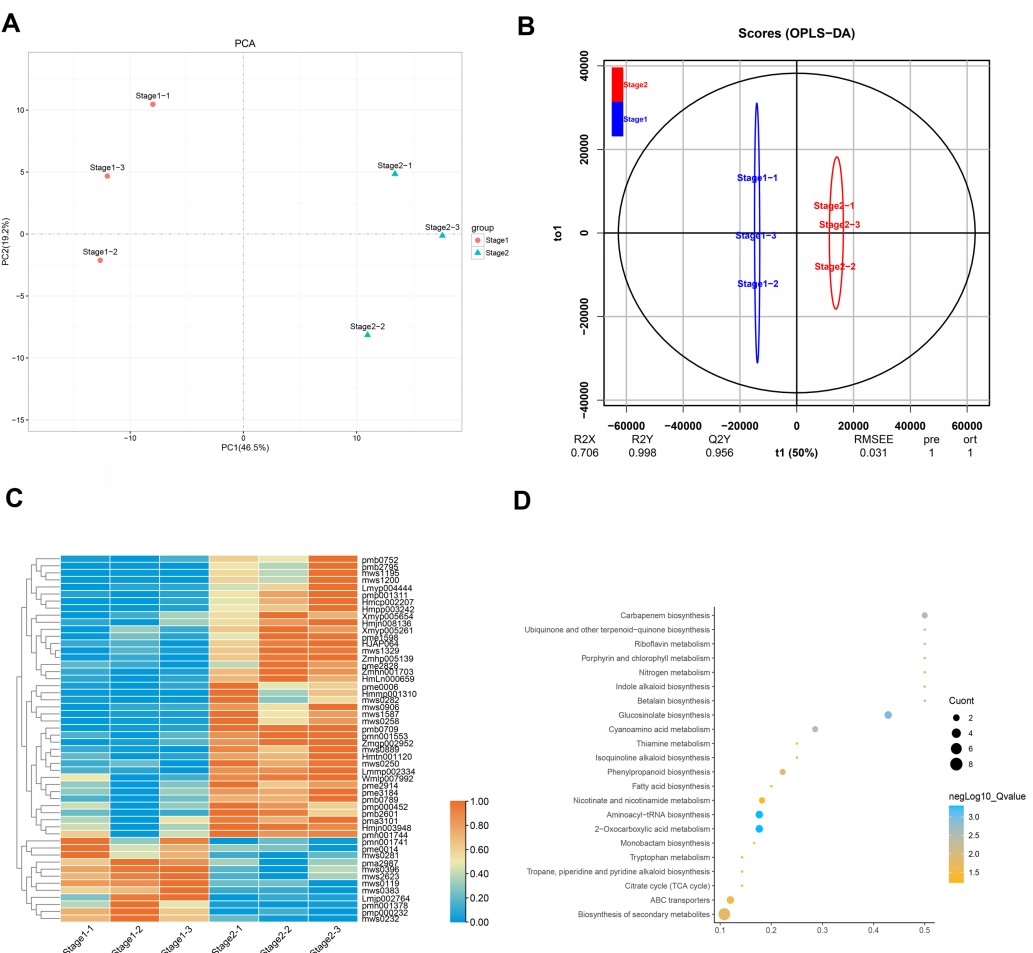

**Figure 5** **Analysis of leaf metabolites in different periods.** (A) Differential metabolite analysis on the basis of principal component analysis (PCA) between stage 1 and stage 2. (B) OPLS-DA model plots for the comparison group stage 1 *vs* stage 2. (C) Differential metabolite heatmap of stage 1 *vs* stage 2, metabolites ID can be found in Table S2. (D) Analysis of common KEGG metabolic pathways between stage 1 and stage 2.

and metabolites to analyze the contribution of each variable (metabolites, genes) to the differences between groups based on the O2PLS model analysis. The loading value represents the explanatory power of the variables (metabolites, genes) in each component (the contribution to the difference between groups), and the positive or negative value of the loading value indicates a positive or negative correlation with another group. The larger the absolute value of the load value, the stronger the correlation. By comparing and analyzing the correlation between differential genes and metabolites in the two periods, we found that they were located at both ends of the ordinate; that is, the absolute values were very large (Fig. 2), indicating that the data in this part were highly correlated, which provided a reliable basis for further analysis. Through the analysis of the most relevant top 25 results in the association analysis results, we found that the metabolites in stage 1 (green leaf stage) and stage 2 (red-purple leaf stage) were amino acids and their derivatives, sesquiterpenes,
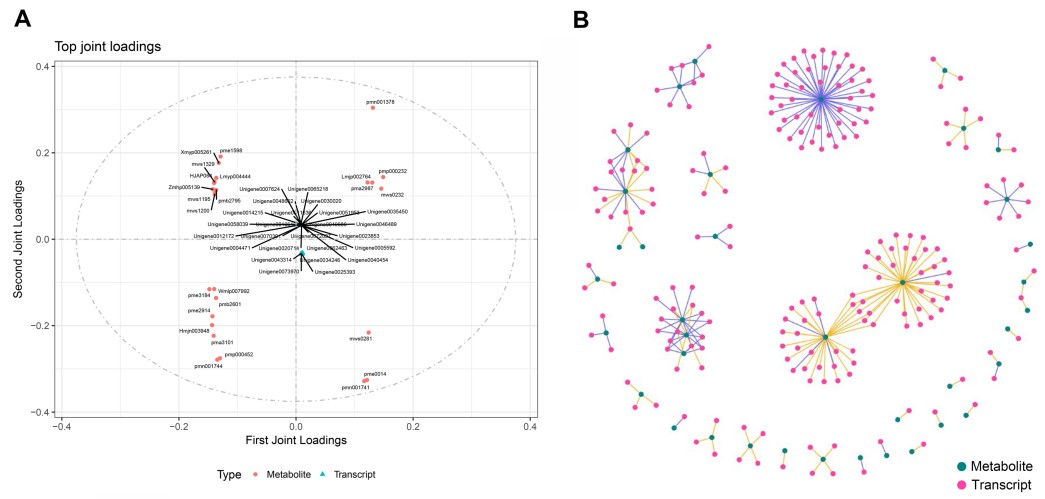

**Figure 6** **Association analysis of metabolomics and transcriptomics data.** (A) Divergent gene and metabolite loading map of stage 1 and stage 2. (B) Network diagram of the correlation between differential gene expression and differential metabolite abundance.

dihydroflavonols, flavonoids, flavonols, triterpenoids, vitamins, coumarins, phenolic acids, lignans, and organic acids, respectively. The genes associated with them were MYB transcription factors, DNA glycosylase superfamily proteins, NAC domain-containing proteins, *etc.* (Fig. 6A).

Through the network association analysis of genes or metabolites in important association positions, we screened out the key associated genes and metabolites, and focused on the analysis of the results of the top 250 (Fig. 6B). The core metabolites were flavonols (Lmmp002334, Lmyp004444, Hmpp003242), alkaloids (pmp001275), free fatty acids (pmp001276), phenolic acids (mws0885, mws0183), flavonoids (Hmpp003270), amino acids and their derivatives (pme0193), and organic acids (mws0639). The genes related to them were various transcription factors (*MYB, NAC, bHLH, WRKY, TCP, etc.*) and various enzymes.

### Screening of anthocyanin synthesis-related genes

Based on the above transcriptome and metabolomics data analysis results, we found that differential genes or substances were mainly concentrated in the anthocyanin synthesis pathway. Therefore, we analyzed the genes related to the anthocyanin synthesis pathway. Through the analysis of expression differences, we screened seven genes with higher expression levels in the red and purple leaf stage of the leaves of *Fraxinus angustifolia* (Figs. 7A and 7B), which were *Unigene0029203* (*4CL3*), *Unigene0037675* (*PAL*), *Unigene0061770* (*DFRA*), *Unigene0009106* (*CHI*), *Unigene0010118* (*PAL*), *Unigene0037674* (*PALA*), and *Unigene0002804* (*CHI3*). The significant differential expression of these genes in two different periods may be closely related to the change of leaf color.

Anthocyanins are not stable in plant cells, and they are generally modified (methylation, acylation, glycosylation, *etc.*) to form more stable and richer anthocyanidin, which

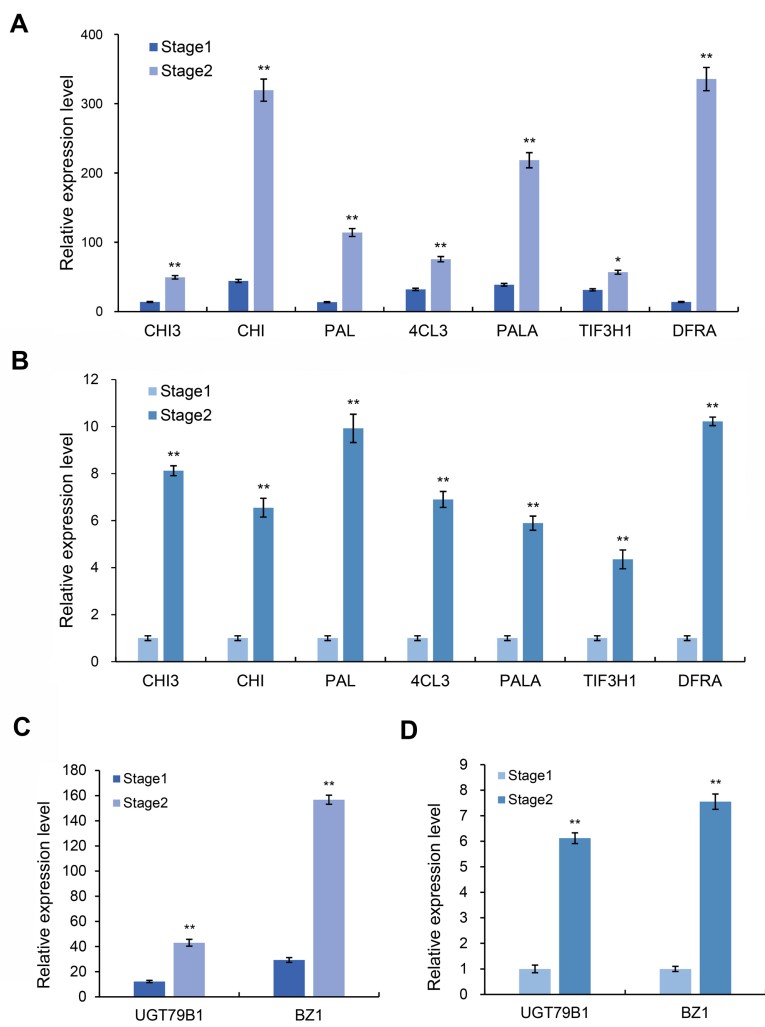

**Figure 7** **Expression level of anthocyanin and synthesis pathway-related genes.** (A) Transcriptome data of anthocyanin synthesis pathway-related genes. (B) qRT-PCR results of anthocyanin synthesis pathway-related genes in different stages. Error bar represents the mean ±SD of three independent biological replicates. (C) Transcriptome data of anthocyanidin synthesis pathway-related genes *BZ1* and *UGT79B1*. (D) qRT-PCR results of *BZ1* and *UGT79B1* in two stages. Error bar represents the mean ±SD of three independent biological replicates.

directly determine the color of plant tissues. We further analyzed the genes related to the anthocyanidin biosynthesis pathway in the transcriptome data and found that two Unigenes in the anthocyanidin biosynthesis pathway were upregulated at the red-purple leaf stage (Figs. 7C and 7D), one of which was *Bronze-1* (*BZ1, anthocyanidin 3-O-glucosyltransferase gene*) and the other gene was *UGT79B1* (*anthocyanidin 3-O-glucoside 2ı-O-xylosyltransferase gene*) (Fig. S4). These two genes showed significant expression differences in different samples, indicating that they may play an important role in leaf color change.

## DISCUSSION

Environmental changes can lead to changes in the color of plant leaves. Usually, changes in external conditions such as temperature, light, humidity, stress environment, reproductive growth, and other external conditions can induce plant leaves to produce corresponding signals, thereby regulating the expression of leaf color change-related genes and the occurrence of related physiological processes, so that leaves show different colors (*Sork et al., 2010*). Studying the physiological and molecular regulation mechanism of leaf discoloration of color-leafed plants, exploring the related genes involved in leaf color regulation, and clarifying their functions and mechanisms will lay a solid theoretical foundation for enriching the landscape and provide important theoretical guidance for the application and promotion of color-leafed tree species (*Zhao et al., 2020*).

Pigments in plants mainly include chlorophyll, carotenoids, flavonoids, and anthocyanins (*Zhao et al., 2020*). Red is mainly derived from the composition and content of flavonoids. Flavonoids are mainly anthocyanins and belong to the more important secondary metabolites, mainly cyanidin aglycone, delphinidin aglycone, pelargonidin aglycone, peoniflorin aglycone, petunidin, and malvidin 6 anthocyanins derived from aglycone. When the anthocyanin content is high, the leaves appear red (*Kong et al., 2003*). New red leaves of tree peony are due to increased anthocyanin content and decreased chlorophyll and carotenoid content (*Luo et al., 2017*). The anthocyanin content in young leaves of jatropha is 11 times the chlorophyll content, mostly in epidermal cells, making leaves red (*Ranjan et al., 2014*). In this study, we performed transcriptome analysis of the leaves of *Fraxinus angustifolia* and found that 5,827 genes were differentially expressed at two different developmental stages. GO enrichment analysis showed that the differentially expressed genes in the three periods were significantly enriched in flavonoid anabolism, phenylpropanoid anabolism, chlorophyll quinone metabolism, pigment metabolism, carotene metabolism, polysaccharide metabolism, terpenoid anabolism, jasmonic acid metabolism, fatty acid anabolism, photosynthesis, secondary metabolite biosynthesis, pigment accumulation, and other terms, indicating that the differentially expressed genes enriched in these terms may play an important role in leaf color change of *Fraxinus angustifolia*.

At the same time, KEGG metabolic pathway enrichment analysis showed that the differential genes were significantly enriched in the phenylpropanoid biosynthesis pathway, which is responsible for the anabolism of flavonoids, such as anthocyanins, flavonols, and flavanols (*Liu et al., 2021*). In many plants, the flavonoid pathway exists in colored tissues and is the main cause of color changes, such as *Tulipa gesneriana* (*Guo et al., 2022*), *Prunus persica* (*Ravaglia et al., 2013*), *etc*. Through gene function annotation, it is valuable to understand the accumulation of anthocyanins, and it is also important for further study of functional genes (*Santos-Buelga, Mateus & De Freitas, 2014*; *Belwal et al., 2020*). In this study, KEGG enrichment analysis identified two important pathways related to anthocyanin biosynthetic metabolism; the flavonoid biosynthesis pathway, flavonoid and flavonol biosynthesis pathways, and anthocyanin metabolic pathways related to

anthocyanin substrate synthesis. Therefore, these genes may have a direct correlation with the synthesis of anthocyanin in *Fraxinus angustifolia*.

Plant metabolomics enables qualitative and quantitative analysis of small molecule metabolites and understanding of their synthesis and metabolic patterns (*Schauer & Fernie, 2006*; *Wang & Kadarmideen, 2020*). At present, the results show that the study of plant metabolites is important for the identification of metabolites, variety differentiation and molecular breeding (*Rizzato et al., 2017*; *Zhou et al., 2017*). The formation of plant color is closely related to the anabolism of flavonoids (*Zhang, Liu & Ruan, 2017*). In the metabolomic analysis of flavonoids in white and purple flowers of Phalaenopsis, 142 different flavonoid-related metabolites were identified, proving that the major anthocyanin is a cyanide derivative (*Meng et al., 2020*). In this study, we determined the metabolites of the leaves of *Fraxinus angustifolia*, and OPLS-DA analysis showed that flavonols were the main factors for the difference of leaf color in different periods of *Fraxinus angustifolia*, and 118 compounds were identified. It is speculated that flavonoid compounds may be the main factor causing leaf color changes. The metabolic pathway enrichment analysis of differential metabolites showed that the metabolic pathways related to anthocyanins synthesis were significantly enriched, which provided an important basis for downstream analysis.

It has been confirmed in plants that transcription factor families, such as MYB and bHLH, are essential for inducing anthocyanin biosynthesis pathways (*Allan, Hellens & Laing, 2008*; *Tanaka, Sasaki & Ohmiya, 2008*). For example, mutations in these regulatory genes usually lead to color changes in apple, *Actinidia chinensis* purple cauliflower (*Brassica oleracea* L.), *Dahlia pinnata,* and tomatoes (*Butelli et al., 2008*; *Chiu et al., 2010*; *Ohno et al., 2011*; *Wang et al., 2019*; *Sun et al., 2021*). We further performed correlation analysis of transcriptome and metabolome data, and we found that there were multiple common metabolic pathways in stage 1 (green leaf stage) and stage 2 (red-purple leaf stage). Based on the O2PLS model analysis, we found that the flavonoid compounds in the substances closely related to the genes in two different periods had a high proportion and were at the core association position of the metabolite network diagram. We found that flavonoid compounds are closely related to transcription factors, such as phenolic acids (mws0183, mws0885, mws1195, mws1200, and pmb2795), flavonoids (HJAP064, HJAP065, Hmpp003270, Xmyp005261, and Zmhp005139), and flavonols (Hmpp003242, Lmdp003808, Lmmp002334, Lmyp004444, and pmp001311). Association analysis results showed that more than 200 genes were closely related to these metabolites, among which transcription factors such as MYB, bHLH, and NAC were closely related to these metabolites. Therefore, these results indicated that transcription factors play an important role in the change of flavonoid compounds during the change of leaf color in autumn. In addition, we also screened nine differentially expressed genes related to anthocyanins. Transcriptome data and qRT-PCR results showed that these genes had significant expression differences in samples of different periods of *Fraxinus angustifolia*. We predicted that they were the key structural genes of anthocyanin biosynthesis and the main factors regulating leaf color changes, which provided an important basis for further analysis of the molecular regulation mechanism of leaf coloration of *Fraxinus angustifolia*.

### Funding

This work was supported by the Key Project of Education Department of Hebei Province (ZD2022009), the National Key Research and Development Project (2020YFD1000700), and the Funding Program for Visiting Scholar Abroad (C20200334). The funders had no role in study design, data collection and analysis, decision to publish, or preparation of the manuscript.

### Grant Disclosures

The following grant information was disclosed by the authors:
Key project of Education Department of Hebei Province: ZD2022009.
National Key Research and Development Project: 2020YFD1000700.
Funding Program for Visiting Scholar Abroad: C20200334.

### Competing Interests

The authors declare there are no competing interests.

### Author Contributions

- Yanlong Wang performed the experiments, analyzed the data, prepared figures and/or tables, authored or reviewed drafts of the article, and approved the final draft.
- Jinpeng Zhen performed the experiments, analyzed the data, prepared figures and/or tables, authored or reviewed drafts of the article, and approved the final draft.
- Xiaoyu Che performed the experiments, analyzed the data, prepared figures and/or tables, authored or reviewed drafts of the article, and approved the final draft.
- Kang Zhang conceived and designed the experiments, prepared figures and/or tables, authored or reviewed drafts of the article, and approved the final draft.
- Guowei Zhang analyzed the data, authored or reviewed drafts of the article, and approved the final draft.
- Huijuan Yang analyzed the data, authored or reviewed drafts of the article, and approved the final draft.
- Jing Wen analyzed the data, authored or reviewed drafts of the article, and approved the final draft.
- Jinxin Wang analyzed the data, authored or reviewed drafts of the article, and approved the final draft.
- Jiming Wang analyzed the data, authored or reviewed drafts of the article, and approved the final draft.
- Bo He analyzed the data, authored or reviewed drafts of the article, and approved the final draft.
- Ailong Yu analyzed the data, prepared figures and/or tables, authored or reviewed drafts of the article, and approved the final draft.
- Yanhui Li conceived and designed the experiments, prepared figures and/or tables, authored or reviewed drafts of the article, and approved the final draft.

- Zhigang Wang conceived and designed the experiments, prepared figures and/or tables, authored or reviewed drafts of the article, and approved the final draft.

## Data Availability

The transcript assembly data generated in this study are available in the National Genomics Data Center (NGDC): PRJCA014072.

## Supplemental Information

Supplemental information for this article can be found online at http://dx.doi.org/10.7717/peerj.15319#supplemental-information.

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
