# Peer review of "Transcriptomic and metabolomic analysis of autumn leaf color change in Fraxinus angustifolia"

_PeerJ, doi:10.7717/peerj.15319_

## Round 0.1 · original submission · Minor Revisions

Kindly read reviewer comments carefully and revise accordingly.

Reviewer 1 ·

Basic reporting

1. Overall, the manuscript was well-written with clear, professional English language. The introduction and discussion provided references that supported the rationale and interpretation of the findings.

2. Some figures require more details for clarity. Please see comments below.

a. Figure 1. Legends are incorrect. (A) refers to two leaves at different stages. (B) refers to length distribution of Unigenes. (C) refers to Venn diagram

b. In Fig 2C, please indicate what the numbers ranging from -2 to 2 represent.

c. In Figure 5C, what do those words on the right side of the heatmap refer to? E.g. pme0006, Hmmp001310, nws0282 etc. It would help the audience to clarify the names of these metabolites for the audience.

3. In section 3.4 Functional Analysis of Differential Genes, the authors explain the results of gene functional enrichment analysis without referring to Figure 3. Please make correct references to Figure 3.

4. Similarly, the authors start discussing KEGG pathway enrichment analysis in line 252 but makes no reference to Figure 4. Please refer to Figure 4 in the discussion to avoid confusion for the audience.

Experimental design

5. Overall, the experimental design was scientifically and logically sound, and the authors provided explanations for why they chose each method for analysis, which was helpful.

6. Please provide more details on how mass spectrometry analysis was performed. For instance, the LC gradient information was omitted. What parameters were used for the setting of the instrument? In addition, did the authors perform any other experiments to confirm the ID of the metabolites? An example would be comparing the data to authentic standards.

Validity of the findings

7. The combination of transcriptomic and metabolomic analysis was able to identify genes whose expression level changes between stage 1 and stage 2, and these findings were further confirmed by qRT-PCR. I commend the authors for going further to confirm their findings from the transcriptomic and metabolomic analysis, which strengthen their findings.

8. In lines 230-233, the authors comment that the separation of stage 1 and stage 2 principal components indicate that 'the sample repeatability was good'. In addition to this, the PCA analysis indicates that the stage 1 and stage 2 leaves contain distinct transcriptomic signatures when all the transcriptomes are considered.

9. As pointed out in point 6, it is not clear whether the authors took additional steps to make sure that the ID of the metabolites identified is accurate. If such additional steps were not taken, it is important to point out in the manuscript that the IDs are speculative, which would be a limitation of the study.

Additional comments

On balance, the manuscript was well-written, and if the points mentioned above are addressed by the authors properly, I suggest this manuscript be accepted with minor modifications.

Annotated reviews are not available for download in order to protect the identity of reviewers who chose to remain anonymous.

Reviewer 2 ·

Basic reporting

The manuscript by Wang et.al. identified the genes and metabolites involved in Fraxinus angustifolia leaf color change through transcriptomic and metabolomic analysis. The study provided a comprehensive profile, which would be helpful for future plant breeding and/or pigment pathway research.

Below are a few minor comments. Hope the authors could address before considered for publication.
1. Please provide more information on sample collection and conditions. I.e. the date/temperate/day length differences when the 2 stage leaf samples were collected; did the replicate come from the same tree or different ones? The specific age of the trees when samples were collected (the pigment formation process might change in different stages of life span).
2. Please update the legends for fig.1 and fig.7, where the descriptions did not match subplots.
3. The association analysis between genes and metabolites was difficult to understand. Could the authors provide more information, i.e. did the analysis involved only differential genes/metabolites or the entire transcriptome/metabolome? Any orthogonal method to validate the predicted association? Any connection or pattern between the highly associated genes and metabolites?
4. Please add detailed description and annotation for fig.6, i.e. what was the meaning of each axis and distance in 6A? what were the names of the metabolites in the clusters in 6B? what were the association strength/significance and direction?
5. It would be helpful if the authors could show representative flavonoid/anthocyanin biosynthesis/metabolism diagrams, and display the corresponding genes and metabolites that were differentially expressed/accumulated in green and red leaves, which would give a clear overview of how the pathways were regulated during pigment formation.

Experimental design

no comment

Validity of the findings

no comment

Reviewer 3 ·

Basic reporting

In this manuscript, Wang et al. investigated the transcriptomic and metabolomic profiles of autumn leaves in Fraxinus angustifolia. The color of autumn leaves differs between stage 1 (green leaf ) and stage 2 (red-purple leaf). The DEGs are involved in several pigment metabolism. Interestingly, they found that flavonoid compounds were different and the nine genes related to anthocyanins were also significantly differentially expressed. The results and methods are clear and convincing. The manuscript is well-written.

Experimental design

The research question is well-defined, relevant and meaningful.

Validity of the findings

The manuscript has an important guiding significance for directing the breeding of colored-leaf Fraxinus species.

Additional comments

I only have some minor suggestions for improvements.
Minor comments:
1. Figure 2, please correct the labels.
2. Figure 7. Have authors conducted statistics for the results? How many biological replicates are used in those results?

---

## Round 0.2 · Minor Revisions

The manuscrit needs minor revision. Please address all the comments carefully

Reviewer 1 ·

Basic reporting

After revision and incorporating additional materials and date requested by the reviewers, the manuscript is structured in a more scientifically and logically sound way. I thank the authors for providing the information that was requested with accuracy. There are some minor points:

1. In Fig 2D, please mention in the figure title that the scores ranging from -2 to 2 represent Z-scores. This will help the audience.

2. In Fig 5C, please mention that the ID of the metabolites can be found in table S2. This will help the audience.

3. In Table S2, please comment on what the values in columns H through M represent.

Experimental design

Thank you for providing additional details for ow mass spectrometry analysis was performed, The information on LC gradient and other instrument parameters is crucial for reproducibility and for the audience

Validity of the findings

The authors provided sufficient data to validate their findings.

Additional comments

Overall, the authors provided more details that were requested by the reviewers, and revisions were made accordingly. After modifying the points mentioned above, I suggest the manuscript be accepted.

Reviewer 2 ·

Basic reporting

The authors have addressed the comments very well. No further comments.

Experimental design

no comment

Validity of the findings

no comment

Reviewer 3 ·

Basic reporting

The responses and revise manuscript perfectly answer my concerns and I recommend accepting this well-written manuscript in its current version.

Experimental design

No comment

Validity of the findings

No comment

Additional comments

I think the current version is ready to be accepted. Thanks for the authors' efforts.

---

## Round 0.3 · Minor Revisions

The manuscript is almost ready for publication.

Please address the following points raised by a Section Editor:

1) Remove "comprehensive" from the title and anywhere else in the manuscript. Although I think the study design is fine, it is not a comprehensive study with two stages and only one extraction/column protocol for metabolites. (Really no study is comprehensive)

(2) The methods description of Illumina library construction is super vague: "poly(A) was added, and it was connected to the Illumina sequencing adapter. After PCR amplification..." Please expand.

3) The transcript assemblies need to be deposited in a public repository, not just the raw reads.

---

## Round 0.4 · accepted · Accept

The issues raised are addressed now. The manuscript is ready for publication.